# Assessment of health staff's proficiency and quality of key malaria indicators in rural district of Ghana

Richard Okyere Boadu[1]*, Hor Karimeni Karimu[2], Kwame Adu Okyere Boadu[3], Obed Uwumbornyi Lasim[1], Lady Agyei Boatemaa[4], Solomon Abotiba Atinbire[5], Nathan Kumasenu Mensah[1]

1 Department of Health Information Management School of Allied Health Sciences, College of Health and Allied Health Sciences, University of Cape Coast, Cape Coast, Ghana, 2 Health Information Management Unit, Sissala East Municipal Health Directorate, Ghana Health Service, Tumu, Ghana, 3 School of Medicine and Dentistry, College of Health Sciences, Kwame Nkrumah University of Science and Technology, Kumasi, Ghana, 4 Department of Information Communication Technology Education, Juaben Senior High School, Ghana Education Service, Accra, Ghana, 5 American Leprosy Mission, Accra, Ghana

* richard.boadu@ucc.edu.gh

**Data Availability Statement:** Data may be obtained upon request from the corresponding author and with permission from Sissala East

## Abstract

### Background

Routine Health Information Systems (RHIS) are important for not just sure enough control of malaria, but its elimination as well. If these systems are working, they can extensively provide accurate data on reported malaria cases instead of presenting modelled approximations of malaria burden. Queries are raised on both the quality and use of generated malaria data. Some issues of concern include inaccurate reporting of malaria cases as well as treatment plans, wrongly categorizing malaria cases in registers used to collate data and misplacing data or registers for reporting. This study analyses data quality concerning health staff's proficiency, timeliness, availability and data accuracy in the Sissala East Municipal Health Directorate (MHD).

### Methods

A cross-sectional design was used to collect data from 15 facilities and 50 health staff members who offered clinical related care for malaria cases in the Sissala East MHD from 24th August 2020 to 17th September 2020. Fifteen health facilities were randomly selected from the 56 health facilities in the municipality that were implementing the malarial control programme, and they were included in the study.

### Results

On the question of when did staff receive any training on malaria-related health information management in the past six months prior to the survey, as minimal as 13 out of 50(26%) claimed to have been trained, whereas the majority 37 out of 50 (74%) had no training. In terms of proficiency in malaria indicators (MI), the majority (68% - 82%) of the respondents could not demonstrate the correct calculations of the indicators. Nevertheless, the MHD

Municipal Health Directorate. The MHD could be contacted through email address: sissalaeastdha2018@gmail.com.

**Funding:** The author(s) received no specific funding for this work.

**Competing interests:** The authors have declared that no competing interests exist.

recorded monthly average timeliness of the 5th day [range: 4.7–5.7] within the reporting year. However, the MHD had a worse average performance of 5.4th and 5.7th days in July and September respectively. Furthermore, results indicated that 14 out of 15(93.3%) facilities exceeded the target to accomplish report availability (> = 90%) and data completeness (> = 90%). However, the verification factor (VF) of the overall malaria indicator showed that the MHD neither over-reported nor under-reported actual cases, with the corresponding level of data quality as Good (+/-5%).

## Conclusions

The Majority of staff had not received any training on malaria-related RHIS. Some staff members did not know the correct definitions of some of MI used in the malaria programme, while the majority of them could not demonstrate the correct calculations of MI. Timeliness of reporting was below the target, nevertheless, copies of data that were submitted were available and completed. There should be training, supervision and monitoring to enhance staff proficiency and improve the quality of MI.

## Introduction

Routine Health Information Systems (RHIS) are important for not just sure enough control of malaria, but its elimination as well [1, 2]. If these systems are working, they can extensively provide accurate data on reported malaria cases instead of presenting modelled approximations of malaria burden [3, 4]. These are the importance of data to monitor the headway of malaria control, a champion for sufficient investments, rally behind the right apportioning of resources and surveillance of diseases [5]. Malaria is endemic in many Sub-Saharan African countries, but RHIS are nothing to write home about, they are delicate. In these parts, too, questions are raised on both the quality and use of generated malaria data [6–8]. Despite the feebleness of routine RHIS, the rejuvenated urge towards getting rid of malaria has revivified the fascination with malaria data these systems produce.

The Global Technical Strategy for Malaria 2016–2030 [9] reiterates the importance of adequate investments in both supervision and employment of RHIS data to assist programme planning as well as enforcement and assessment [10, 11]. One goal of the National Malaria Strategy 2019–2023 is to boost malaria surveillance and use the information derived to better the decisions taken for programme performance. Ghana is trying to reflect this worldwide interest [12–14]. The Ghana Health Service (GHS), in its quest to standardize data collation, has come up with registers and forms which are at par for all public and mission health facilities in Ghana. GHS has therefore mandated that these public and mission health facilities use these registers and forms as standard tools to collect and present data. To improve RHIS data quality and encourage the use of data for decision-making, the President's Malaria Initiative (PMI) in collaboration with the Ghana Health Service (GHS) developed the District Health Information Management System (DHIMS) in 2012. This is a dynamic web-based data management system and has contributed remarkably to the nation's RHIS activities, thus, improving data collection, reporting, and analysis. The system makes it possible for health facilities all over the country to enter their routine reports directly into an electronic database [15, 16]. In addition to ensuring continuous RHIS data quality assurance practices (QAPs), the GHS provided standard operating procedures (SOP) on health information which provides a

formalized system to guide data collection, collation, storage, and analysis, reporting and use. The purpose of the SOP is to attain maximum accuracy, completeness, integrity and traceability of all data (including data for malaria indicators) in the GHS and other health programme implementing agencies [17]. The SOP outline the procedure to manage data to ensure completeness, accuracy and timeliness of data to facilitate decision-making in the service. Data collectors and managers should be sufficiently familiar with this SOP [18, 19].

Malaria is a major cause of illnesses and death in Ghana, particularly among children and pregnant women. In 2012, malaria accounted for 38.9% of all out-patient illnesses and 38.8% of all admissions [20]. Malaria infection during pregnancy causes maternal anaemia and placental parasitaemia, both responsible for miscarriages and low birth weight babies. As much as 16.8% of all admissions of pregnant women in 2012 were attributable to malaria. Interestingly, 3.4% of death among pregnant women were also due to malaria. Malaria parasite prevalence among children aged 6–59 months in the 2011 report indicated a regional variation from as low as 4% in the Greater Accra region to as high as 51% in the Upper West region [21]. Malaria remains the number one cause of OPD attendance and the first among the top ten diseases in Ghana [22]. The National Malaria Control Program (NMCP) in collaboration with the Upper West Regional Health Directorate (RHD) and Sissala East Municipal Health Directorate (MHD) continue to implement interventions across the municipality aimed at controlling the burden of the disease within the municipality, region and Ghana as a whole. These interventions include Long-lasting insecticidal nets (LLIN) distribution, In-door Residual Spraying and Seasonal Malaria Chemoprevention (SMC) among others [22].

Since Ghana adopted the Roll Back Malaria Initiative in 1998/1999, the country has been implementing a combination of preventive and curative interventions as outlined in the Strategic Plan for Malaria Control in Ghana, 2014–2020. The country continues to implement strategies that are designed to enhance the attainment of the set objectives. Additionally, Ghana subscribes to sub-regional and global initiatives such as the T3 (Test, Treat and Track) initiative which seeks to ensure that every suspected malaria case is tested, that every case tested positive is treated with the recommended quality-assured antimalarial medicine, and that the disease is tracked through timely and accurate reporting to guide policy and operational decisions [23]. Despite these strides, the quality of malaria data available to managers is perceived to be of poor quality, which is similar to what is characterised in the Sissala East MHD [24]. This has limited the use of malaria data to meet the reporting needs of the municipal managers. Previous research in Ghana and other sub-Saharan African countries has highlighted issues of inaccurate reporting of malaria cases as well as treatment plans, wrongly categorizing malaria cases in registers used to collate data and misplacing data and or registers for reporting [25–28]. While previous attempts to address the issues of Malaria data quality have been concentrated at the national level, discrepancies with collated data against what pertains at district and regional levels exist. Little research has evaluated the contribution of facility level staff proficiency in Malaria data quality to identify potential gaps and to implement any data quality improvement intervention. This analysis intends to contribute to filling this knowledge gap as Ghana moves towards the elimination of Malaria. This study presents findings of data quality with respect to health staff's proficiency, timeliness, availability and data accuracy in the Sissala East MHD.

## Research methods

### Study design

The study used a cross-sectional design to collect data from 15 facilities and 50 health staff who offer clinical related care for malaria cases in the Sissala East MHD from 24 August 2020–17

September 2020. This design sought to determine the proficiency of healthcare professionals on key malaria indicators and the level of data quality in the MHD.

## Study area

The Sissala East Municipality is one of the eleven municipals in the Upper West Region. The Municipality was carved out from the Old Sissala District in 2005 by the government for effective governance and decentralization, making Sissala West (Gwollu) a separate district. Tumu got elevated to the status of a municipality in 2018. The Municipal is bounded to the East by the Upper East Region (Kassena Nankana and Builsa Municipals), South by Wa East, West Mamprusi Municipal and Nadowli Municipal, West by Sissala West Municipal and to the North by the Republic of Burkina Faso. It falls between longitudes 1.300 W and latitudes 10.000 N and 11.000 N. It has a total land size of 4,744 sq. km, representing 26% of the total landmass of the region. The Municipality continues to face inadequate numbers in terms of staff strength. Cadres such as Health Information Officers, Disease Control Officers, Nutrition Officers, Health Promotion Officers, Heath Record Assistants, Community Health Nurses etc. continue to dwindle in numbers due to the high staff attrition rate. These cadres undoubtedly form the core staff with the requisite skill set to ensure that data is at the highest of standards in terms of quality. Table 1 shows the characteristics of the selected facilities such as number of population/coverages, male: female ratio, number of health staff and doctor/nurse: patient ratio. [Note: Projected population was based on Ghana's 2010 population and housing census. Both HOS & HC5 are located with the same catchment area and serve the same population].

## Selection of sites and participants

Sissala East MHD was purposely selected, due to concerns regarding the perceived quality of their reported data. However, only 15 health facilities were randomly selected from the 56 health facilities in the municipality that were implementing the malarial control programme

**Table 1. Characteristics of the facilities included in the study.**

| No. | Facility Name | Population/ Coverage | Male: Female Ratio | No. of Health Staff involved in malaria case management | No. of Health Staff interviewed | Doctor to Patient Ratio | Nurse to Patient Ratio |
|---|---|---|---|---|---|---|---|
| 1 | HOS | 7611 | 9:10 | 29 | 19 | 1: 3805.5 | 1:423 |
| 2 | HC1 | 1294 | 8:12 | 5 | 3 | 0 | 1:259 |
| 3 | HC2 | 425 | 9:10 | 4 | 2 | 0 | 1:106 |
| 4 | HC3 | 1953 | 8:11 | 7 | 4 | 0 | 1:279 |
| 5 | HC4 | 4328 | 8:11 | 5 | 3 | 0 | 1:866 |
| 6 | HC5 | NA | NA | 5 | 3 | NA | NA |
| 7 | HC6 | 896 | 8:11 | 3 | 2 | 0 | 1:299 |
| 8 | HC7 | 5784 | 9:10 | 11 | 6 | 0 | 1:723 |
| 9 | C1 | 943 | 9:10 | 2 | 1 | 0 | 1:472 |
| 10 | C2 | 789 | 9:10 | 2 | 1 | 0 | 1:395 |
| 11 | C3 | 953 | 9:10 | 2 | 1 | 0 | 1:477 |
| 12 | C4 | 1301 | 9:10 | 2 | 1 | 0 | 1:651 |
| 13 | C5 | 1206 | 9:10 | 2 | 1 | 0 | 1:603 |
| 14 | C6 | 1129 | 9:10 | 2 | 1 | 0 | 1:565 |
| 15 | C7 | 2980 | 9:10 | 3 | 2 | 0 | 1:993 |
| | **MHD** | **31,592** | **9:11** | **84** | **50** | **1: 15796** | **1:451** |

Source: Sissala East District Municipal Annual Report (2019).

were included in the study. The selected facilities included seven health centres (HC), seven community-based planning and services (C) and one hospital (Hos). At each facility, all staff who offer clinical related care for malaria cases and consented to participate in the study were included and interviewed.

## Sample size determination

A sample size of 15 was selected from 56 facilities in the MHD using StatCalc function in EpiInfo Software Version 3.01 [Confidence level = 95%, expected frequency = 50%, acceptable margin of error = 21%, design effect = 1, cluster = 1].

## Selection of malaria indicators

A core sets of malaria indicators, cutting across both diagnostic and treatment (such as suspected malaria cases tested, malaria cases tested positive. Depending of the facility either microscopy or RDT or both tests were used for diagnosis and confirmed malaria positive. Other indicators included OPD malaria cases put on ACTs, pregnant women receiving IPTp3, and inpatient malaria deaths), were selected for measures of staff proficiency, Timeliness of reporting, completeness of reports and accuracy of reporting. The selection of these key indicators is based on the most critical services being provided within the malaria programme. These indicators were therefore the focus of the data quality analysis.

## Measurements and data analysis

The DQA grounded in the components of data quality, namely, that programmes and projects need accurate, reliable, precise, complete and timely data reports that managers can use to effectively direct available resources and to evaluate progress toward established goals. In addition, the data must have integrity to be considered credible and should be produced ensuring standards of confidentiality. However, the scope of this study was limited to accuracy (also known as validity), completeness and timeliness. Accurate data are considered correct: the data measure what they are intended to measure. Accurate data minimize errors (e.g. recording or interviewer bias, transcription error, sampling error) to a point of errors being negligible. Completeness means that an information system from which the results are derived is appropriately inclusive: it represents the complete list of eligible persons or units and not just a fraction of the list. Data are timely when they are up-to-date (current), and when the information is available on time. Timeliness is affected by: (1) the rate at which the program's information system is updated, (2) the rate of change of actual program activities, and (3) when the information is actually used or required. Based on these dimensions of data quality, the Measure Evaluation DQA Protocol was adopted and is comprised of two components: (1) assessment of data management and reporting systems and (2) verification of reported data for key indicators at selected sites [28–31].

   The purpose of the Protocol is to assess, on a limited scale, if service delivery and intermediate aggregation sites are collecting and reporting data to measure the audited indicator(s) accurately and on time—and to cross-check the reported results with other data sources. To do this, the DQA determined if a sample of Service Delivery Sites have accurately recorded the activity related to the selected indicator(s) on source documents. It is then traced as data to see if it has been correctly aggregated and/or otherwise manipulated as it is submitted from the Participating facilities through the Health Information Management Unit in the MHD. The data verification exercise was conducted in two stages:

**Table 2. Accuracy key.**

| Verification Factor (VF)—Rating per Malaria Indicator | % of facilities for which source data exactly match reported data | % of facilities that over-report by more than 10% (VF < 0.90) | % of facilities that under-report by more than 10% (VF > 1.10) | |
|---|---|---|---|---|
| Level of Data Accuracy | Very poor VF: >20% | Poor | Moderate | Good |
| | | VF: +/-11% to 20% | VF: + /- 6% to 10% | VF: +/-5% |

Source: [32].

- In-depth verifications at the 15 Service Delivery Sites—by selecting key malaria indicators and recounting comparing the data elements contained in the primary records of selected health facilities for the period January 2019 –December 2019.

- Follow-up verifications aggregation (summary report) submitted to the Health Information Management (HIM)/ M&E Unit.

The Verification Factor (VF) is the key metric for assessing the quality of the reported data, by comparing the reported data to the source data (i.e., the register or other HMIS record at the service delivery point). The interpretation of VF and the level of data accuracy is shown in Table 2.

$$\text{Verification Factor (VF)} = \frac{\textit{Rcounted number of events from source documents}}{\text{Reported number of events submitted to HIM Unit}} \text{X } 100$$

The following measurements were used to calculate % of all malaria reports that were A) available; B) on time; and C) complete.

$$\text{A) } \% \text{ Available Reports (available to the Audit Team)} = \frac{\text{Number of reports } \textit{received} \text{ from all participating facilities}}{\text{Number of reports } \textit{expected} \text{ all participating facilties}} \text{X } 100$$

$$\text{B) } \% \text{ On Time Reports (received by the due date)} = \frac{\text{Number of reports } \textit{received} \text{ on time from all participating facilities}}{\text{Number of reports } \textit{expected} \text{ from all participating facilties}} \text{X } 100$$

$$\text{C) } \% \text{ Complete Report} = \frac{\text{Number of reports } \textit{that are complete} \text{ from all participating facilities}}{\text{Number of reports } \textit{expected} \text{ from all participating facilties}} \text{X } 100$$

That is to say, for a report to be considered complete it should include at least (1) the reported count relevant to the indicator; (2) the reporting period; (3) the date of submission of the report; and (4) a signature from the staff having submitted the report.

Health Staff understanding and proficiency in malaria indicator was also measured by a pencil and paper test that measured the ability of respondents to perform calculations, and to interpret and use RHIS malaria results as stipulated in the GHS SOP.

## Ethical considerations

Ethical approval was obtained from the Cape Coast Teaching Hospital Ethical Review Committee (approval number: CCTHERC/EC/2020/045). An introductory letter from the Department of Health Information Management, University of Cape Coast was sent to the facility. The letter explained the purpose of the study as well as the reason for the collecting the data. An approval was given by the MHD. Respondents were assured of confidentiality of the information they would be providing. The purpose of the study and the various sections of the questionnaires were explained to respondents to enable them answer the questions conveniently.

## Limitations

There were instances where the consulting room register did not contain data on the full year, also the other months in a different register could not be traced. This was witnessed in two facilities, this undoubtedly posed a challenge to the availability of source documents to enable the investigators review data as intended. Nonetheless, such situations were treated as missing data with VF interpretation as 'over reported'. Most of the facilities visited did not have a dispensary register where documentation of drugs issued to clients especially ACTs could be verified.

## Results

Majority of the respondents interviewed were females 27(54%) and the rest were males. The mean age of the respondents was 30.8 years (range: 25–40 years). Majority of the respondents were Nurses/Midwives (NM) 21(42%) followed by Community Health Nurses (CHN) 11 (22%) and Technical Officers (TO) 7(14.0%). Doctors and Physician Assistants (PA) contributed to 1(2%) of the respondents respectively. In terms of educational level, majority of the respondents were certificate holders 26(52%), followed by diploma 20(40%) [Table 3].

**Table 3. Socio-demographic characteristics of respondents.**

| Variable | Frequency (n = 50) | Percent (%) |
|---|---|---|
| **Gender** | | |
| Female | 27 | 54 |
| Male | 23 | 46 |
| **Category of Staff** | | |
| Nurse/ Midwife | 21 | 42 |
| Community Health Nurse | 11 | 22 |
| Technical Officer (Disease Control Officer/Nutrition Officer /Field Technician) | 7 | 14 |
| Biostatistician / Health Information Officer. | 4 | 8 |
| Lab scientist/Technician | 3 | 6 |
| Doctor | 1 | 2 |
| Pharmacist | 1 | 2 |
| Physician Assistant | 1 | 2 |
| Dispensary Technician | 1 | 2 |
| **Educational Level** | | |
| Certificate | 26 | 52 |
| Diploma | 20 | 40 |
| Bachelors | 3 | 6 |
| Masters | 1 | 2 |

Source: Survey, 2021.

## Staff proficiency in malaria indicators

Only thirteen (26%) out of fifty respondents had received training on malaria related health information management in the past six months prior to the survey.

Respondents did not understand correctly definitions of common indicators as used in the malaria programme. For instance, the number of pregnant women receiving IPTp3 27(54%) was the most misconstrued indicator, compared to the indicator, number of suspected malaria cases tested 13(26%). With respect to proficiency, the majority of the respondents could not demonstrate correctly, the calculation of the number of pregnant women receiving IPTp3 41 (82%). The number of respondents who lacked proficiency in the calculations of the number of suspected malaria cases and the number of malaria cases tested positive were 34(68%) respectively (Fig 1).

## Understanding and proficiency in malaria indicators by staff category

Eighteen (36%) out of the 50 respondents did not demonstrate understanding of the definition of the indicator "suspected malaria cases tested". Majority of these category of respondents were NM 25(50%) and CHN 14(28%). Of the 50 respondents, 34 of them did not know how to calculate correctly, "suspected malaria cases".

Again, NM and CHN constituted 13(26%) of respondents who did not know how to explain the indicator "number of suspected malaria cases tested positive". About 34 of the respondents comprising of NM 18(53%), CHN 9(26%) and other staff 7(21%) respectively did not know how this indicator was computed (not shown in the table/figure).

With respect to the indicator "number of OPD malaria cases put on ACTs", when the meaning of this indicator was asked, 25(50%) of the respondents had no knowledge about it, of these, 15(60%) were NM and 7(28%) were CHN and the remaining belong to other categories of staff.

Furthermore, 37(74%) of respondents did not know how to calculate "the number of OPD malaria cases put on ACTs". Once again, NM 20(54%) and CHN 9(24%) comprised majority of these category, while the rest were from the other professions (not shown in the table/figure).

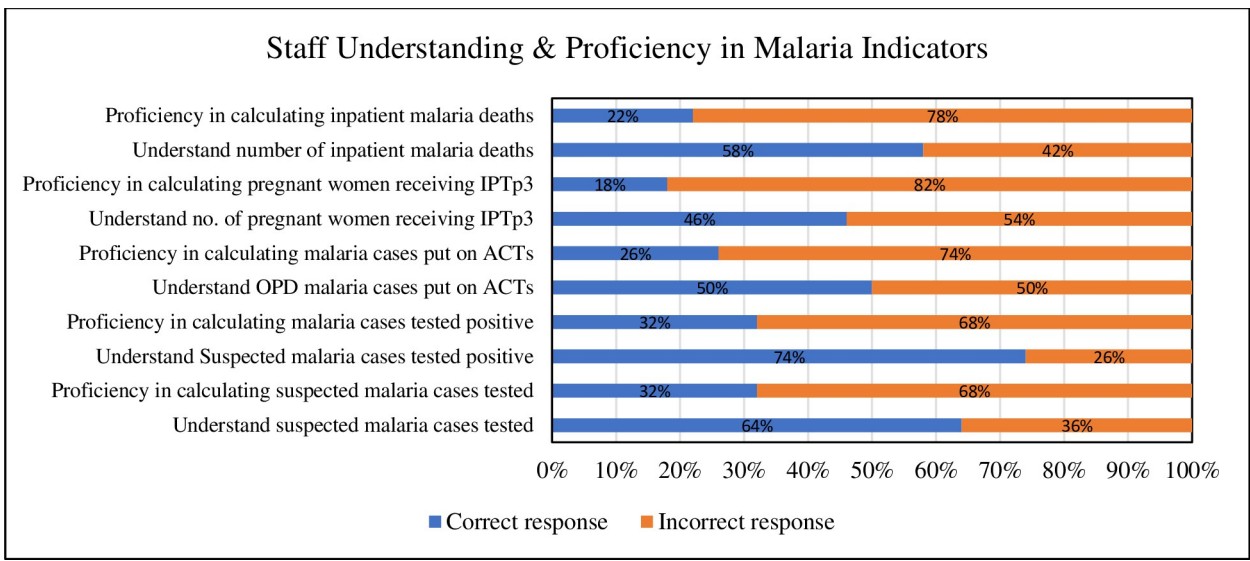

**Fig 1. Staff understanding & proficiency in malaria indicators.** Source: Survey, 2021.

Another vital indicator asked was "the number of pregnant women receiving IPT$_p$3". Only 23(46%) respondents knew the meaning of this indicator. Again, NM 16(59%) and CHN 6 (22%) constituted the majority who did not know and the remaining were from other categories of staff. The health staff understanding of how to calculate this indicator was the worst. Overwhelming majority 41(82%) of the respondents could not calculate it. Yet again, NM constituted the majority with 20 (49%), followed by CHN 11(27%). The remaining 10(24%) are from the other categories of staff (not shown in the table/figure).

The last of the key indicators considered was the number of inpatient malaria deaths. Of the 21(42%) respondents who could not explain it, NM constituted 13(62%), Laboratory Technologist/Technicians 3(14%) and Field Technicians (9%). CHN were the least amongst them (4.3%). When it came to demonstrating understanding of how to calculate this indicator, 39 (78%) of the respondents had no idea about how it was done. NM formed 19 (49%) of these and 11(28%) were CHN, while the other staff constituted 9(23%) (not shown in the table/figure).

## Timeliness of routine malaria data

The timeliness of submission of routine malaria data from the facility to the MHD is the 5th day of the ensuing month, with early bird and late submissions on the 3rd and 8th day respectively. On average, 9 out of 15 (60%) facilities were able to submit their reports on time (i.e. before 5th day of the ensuing month). The MHD recorded a monthly average timeliness of 5th day [range: 4.7–5.7] within the reporting year. However, the MHD had a worse average performance of 5.4th and 5.7th day in July and September respectively (Fig 2).

## Availability of routine malaria data

Ghana Health Service (GHS) standard monthly malaria summary report contains 2,070 data elements (24,840 per year). To achieve 100% completeness implies all the data elements should be completed. Also, to achieve 100% report availability means all reports (12 per year) within the reporting period should be available and accessible with the reporting year. As shown in Fig 3, 14 out of 15(93.3%) facilities exceeded the target to accomplish report availability (> = 90%) and data completeness (> = 90%).

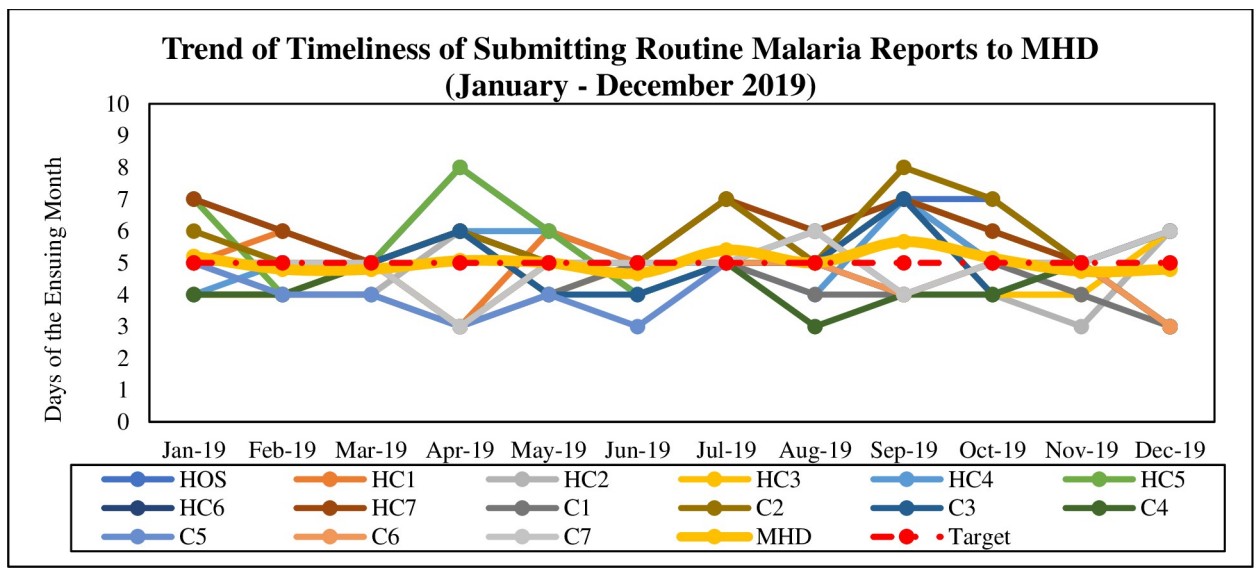

**Fig 2. Timeliness of submitting routine malaria reports.** Source: Survey, 2021.

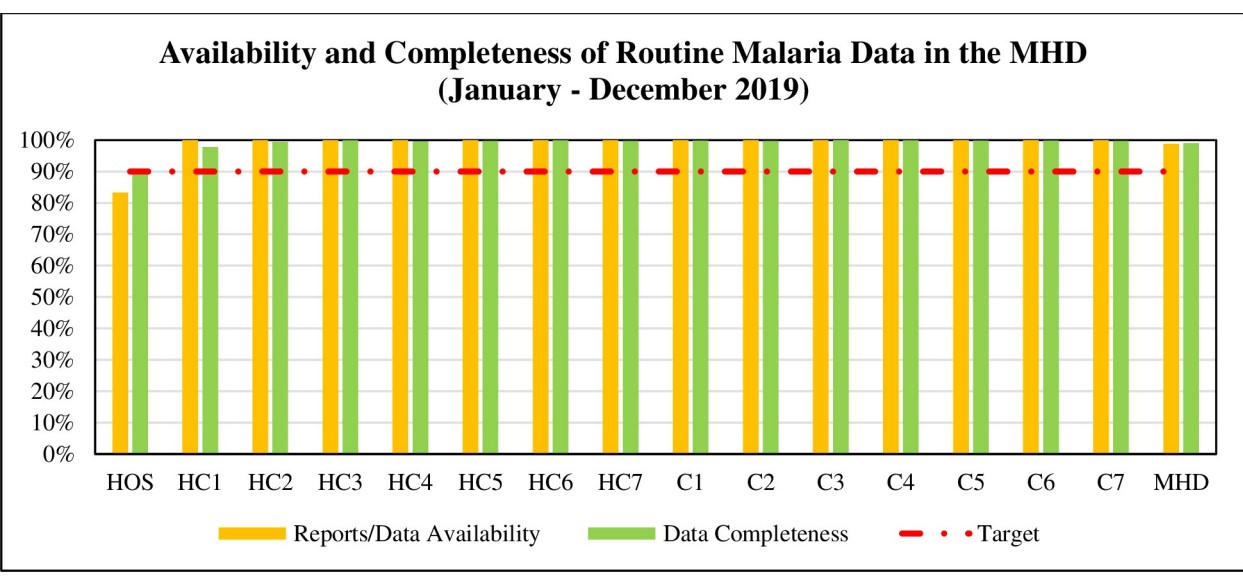

**Fig 3. Availability and completeness of routine malaria data.** Source: Survey, 2021.

### Accuracy of malaria data

From Table 4, 7 out of 15(46.7%) facilities reported the exact number of suspected malaria cases (clinically diagnosed) recorded in the Lab and Consulting Registers during the period of investigation. Another 5 out of 15(33.3%) of the facilities neither over-reported nor under-reported with their corresponding level of data quality as Good (+/-5%). However, the overall verification (VF) indicated the MHD neither over-reported nor under-reported actual cases with corresponding level of data quality as Good (+/-5%).

Table 5 illustrates that, 9 out of 15(60.0%) facilities reported the exact number of suspected malaria cases recorded during the period of the study. Additional 3 out of 15(20.0%) of the

**Table 4. Accuracy of reported number of suspected malaria cases: Verification factor.**

| No. | Facility Name | Recounted Data | Report Data | Verification Factor (VF) | VF Interpretation | Level of Data Quality |
|---|---|---|---|---|---|---|
| | | (A) | (B) | = (A/B) | | |
| 1 | HOS | 16488 | 16720 | 0.99 | No over report or under report | Good |
| 2 | HC1 | 2177 | 2171 | 1.00 | Exact match | Good |
| 3 | HC2 | 1020 | 1019 | 1.00 | Exact match | Good |
| 4 | HC3 | 3811 | 3842 | 0.99 | No over report or under report | Good |
| 5 | HC4 | 1855 | 2343 | 0.79 | Over reported | Poor |
| 6 | HC5 | 380 | 483 | 0.79 | Over reported | Poor |
| 7 | HC6 | 226 | 226 | 1.00 | Exact match | Good |
| 8 | HC7 | 4713 | 4718 | 1.00 | Exact match | Good |
| 9 | C1 | 577 | 579 | 1.00 | Exact match | Good |
| 10 | C2 | 109 | 109 | 1.00 | Exact match | Good |
| 11 | C3 | 576 | 577 | 1.00 | Exact match | Good |
| 12 | C4 | 420 | 426 | 0.99 | No over report or under report | Good |
| 13 | C5 | 149 | 174 | 0.86 | Over reported | Poor |
| 14 | C6 | 721 | 733 | 0.98 | No over report or under report | Good |
| 15 | C7 | 621 | 617 | 1.01 | No over report or under report | Good |
| | **MHD** | **33843** | **34737** | **0.97** | No over report or under report | Good |

**Table 5. Accuracy of reported number of suspected malaria cases tested: Verification factor.**

| No. | Facility Name | Recounted Data | Report Data | Verification Factor (VF) | VF Interpretation | Level of Data Quality |
|-----|---------------|----------------|-------------|--------------------------|-------------------|------------------------|
| | | (A) | (B) | = (A/B) | | |
| 1 | HOS | 16672 | 16724 | 1.00 | Exact match | Good |
| 2 | HC1 | 2174 | 2171 | 1.00 | Exact match | Good |
| 3 | HC2 | 1020 | 861 | 1.18 | Under reported | Poor |
| 4 | HC3 | 3842 | 3499 | 1.10 | No over report or under report | Moderate |
| 5 | HC4 | 1681 | 1806 | 0.93 | No over report or under report | Moderate |
| 6 | HC5 | 380 | 483 | 0.79 | Over reported | Poor |
| 7 | HC6 | 226 | 226 | 1.00 | Exact match | Good |
| 8 | HC7 | 4713 | 4700 | 1.00 | Exact match | Good |
| 9 | C1 | 578 | 579 | 1.00 | Exact match | Good |
| 10 | C2 | 109 | 109 | 1.00 | Exact match | Good |
| 11 | C3 | 579 | 577 | 1.00 | Exact match | Good |
| 12 | C4 | 397 | 426 | 0.93 | No over report or under report | Moderate |
| 13 | C5 | 148 | 174 | 0.85 | Over reported | Poor |
| 14 | C6 | 730 | 733 | 1.00 | Exact match | Good |
| 15 | C7 | 612 | 610 | 1.00 | Exact match | Good |
| | **MHD** | **33861** | **33678** | **1.01** | No over report or under report | Good |

facilities did not either over report or under report, with their corresponding level of data quality as Moderate (+/-6% to 10%). Nevertheless, the overall verification (VF) indicates the MHD neither over reported nor under reported actual cases with corresponding level of data quality as Good (+/-5%).

Table 6 shows that, 8 out of 15 (53.3%) facilities reported the exact number of suspected malaria cases that tested positive in the research period. Apparently, 6 out of 15 (40.0%) of the facilities did not either over report or under report with corresponding level of data quality of 5/6 of the facilities as Good (+/-5%) and 1/5 as Moderate (+/-6% to 10%). Yet, the overall

**Table 6. Accuracy of reported number of malaria cases tested positive: Verification factor.**

| No. | Facility Name | Recounted Data | Report Data | Verification Factor (VF) | VF Interpretation | Level of Data Quality |
|-----|---------------|----------------|-------------|--------------------------|-------------------|------------------------|
| | | (A) | (B) | = (A/B) | | |
| 1 | HOS | 3283 | 3257 | 1.01 | No over report or under report | Good |
| 2 | HC1 | 1257 | 1265 | 0.99 | No over report or under report | Good |
| 3 | HC2 | 579 | 579 | 1.00 | Exact match | Good |
| 4 | HC3 | 2025 | 2025 | 1.00 | Exact match | Good |
| 5 | HC4 | 1018 | 1070 | 0.95 | No over report or under report | Good |
| 6 | HC5 | 222 | 272 | 0.82 | Over reported | Poor |
| 7 | HC6 | 142 | 142 | 1.00 | Exact match | Good |
| 8 | HC7 | 3121 | 3120 | 1.00 | Exact match | Good |
| 9 | C1 | 378 | 379 | 1.00 | Exact match | Good |
| 10 | C2 | 87 | 87 | 1.00 | Exact match | Good |
| 11 | C3 | 355 | 356 | 1.00 | Exact match | Good |
| 12 | C4 | 132 | 141 | 0.94 | No over report or under report | Moderate |
| 13 | C5 | 136 | 140 | 0.97 | No over report or under report | Good |
| 14 | C6 | 498 | 498 | 1.00 | Exact match | Good |
| 15 | C7 | 359 | 368 | 0.98 | No over report or under report | Good |
| | MHD | **13592** | **13699** | **0.99** | No over report or under report | Good |

**Table 7. Accuracy of reported number of OPD malaria cases treated with ACTs: Verification factor (facilities with VF = 0, do not have dispensary register where documentation of drugs issued to clients especially ACTs could be verified).**

| No. | Facility Name | Recounted Data (A) | Report Data (B) | Verification Factor (VF) = (A/B) | VF Interpretation | Level of Data Quality |
|-----|---------------|--------------------|-----------------|-----------------------------------|-------------------|-----------------------|
| 1 | HOS | 2165 | 2311 | 0.94 | No over report or under report | Moderate |
| 2 | HC1 | 1112 | 1251 | 0.89 | Over reported | Poor |
| 3 | HC2 | 657 | 735 | 0.89 | Over reported | Poor |
| 4 | HC3 | 1853 | 1988 | 0.93 | No over report or under report | Moderate |
| 5 | HC4 | 1148 | 1533 | 0.75 | Over reported | Poor |
| 6 | HC5 | 0 | 272 | 0.00 | Over reported | Very poor |
| 7 | HC6 | 0 | 142 | 0.00 | Over reported | Very poor |
| 8 | HC7 | 2992 | 3158 | 0.95 | No over report or under report | Good |
| 9 | C1 | 0 | 379 | 0.00 | Over reported | Very poor |
| 10 | C2 | 0 | 87 | 0.00 | Over reported | Very poor |
| 11 | C3 | 0 | 356 | 0.00 | Over reported | Very poor |
| 12 | C4 | 0 | 141 | 0.00 | Over reported | Very poor |
| 13 | C5 | 0 | 140 | 0.00 | Over reported | Very poor |
| 14 | C6 | 0 | 498 | 0.00 | Over reported | Very poor |
| 15 | C7 | 0 | 323 | 0.00 | Over reported | Very poor |
| | **MHD** | **9927** | **13314** | **0.75** | Over reported | Poor |

verification (VF) indicates the MHD did not either over report or under report actual cases with corresponding level of data quality as Good (+/-5%).

Table 7 reveals that, none of the facilities reported the exact number of reported OPD malaria cases treated with ACTs. However, 12 out of 15 (80.0%) of the facilities over reported with corresponding level of data quality as Poor. The remaining 3 out of 15 (20.0%) of the facilities did not either over report or under report with corresponding level of data quality as Moderate and Good. Nevertheless, the overall VF indicates the MHD over reported OPD malaria cases treated with ACTs with corresponding level of data quality as Poor (+/-11% to 20%).

Fig 4 shows the verification factors performance of the four malaria indicators and the overall malaria indicator at the MHD service delivery sites. The MHD VF of each indicator was summative of all "recounted" and "reported" data points from the fifteen selected health facilities. Also, VF of the overall malaria indicator was summative of all "recounted" and "reported" data points for the key indicators (Number of suspected malaria cases, Number of suspected malaria cases tested, Number of malaria cases tested positive, Number of OPD malaria cases treated with ACTs from the fifteen selected health facilities. We can see that there is a wide variation in the accuracy of these indicators. The area marked with red horizontal lines shows a margin of acceptability: plus or minus 10% of 100%, the global standard. However, individual programmes can select their own ranges of acceptability, as deemed appropriate. We also can see that, of the four indicators, "Number of OPD malaria cases treated with ACTs" is outside the acceptable margins. Ideally, we would see no under-reporting or over-reporting of data, with indicators as close to 100% as possible.

## Discussions

### Socio-demographic characteristics of respondents

Our study revealed that majority of the healthcare providers are females. The socio-demographic characteristics of respondents suggests that the MHD is endowed with energetic workforce with diverse professional speciality required to manage uncomplicated malaria cases.

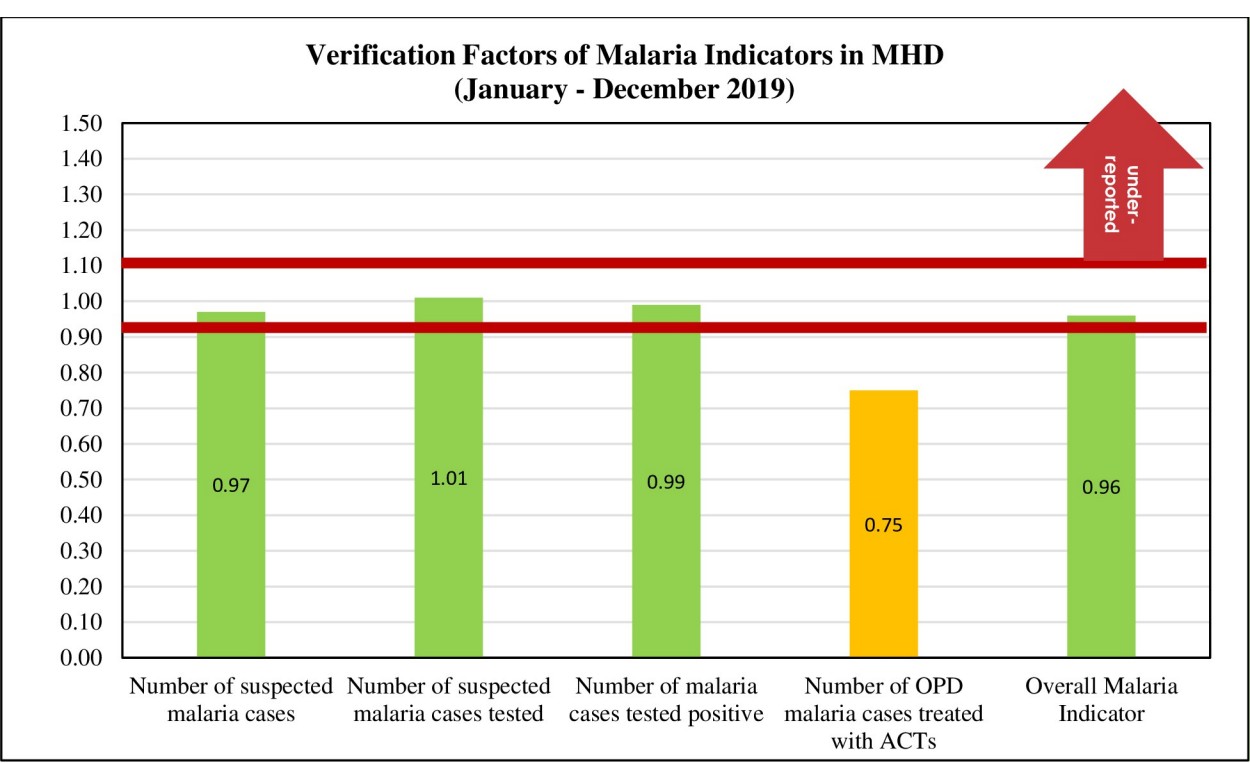

**Fig 4. Verification factors of malaria indicators.** Source: Survey, 2021.

## Staff understanding and proficiency in malaria indicators

In spite of the enviable educational and professional background of the respondents, the MHD seems not to harness this comparative advantage to train its staff on malaria related health information management, as the majority of them (staff) had not received such training in the past six months prior to the study. This to a large extent has quality implications of malaria data produced at all levels of service delivery in the MHD. It was therefore not surprising that some staff members (between 26% and 36%) do not know the correct definitions of some common indicators used in malaria programme, as revealed by other studies [33]. The situation is absurd with respect to definition of OPD malaria cases put on ACTs, as half of respondents had no idea. In terms of proficiency in malaria indicators, majority (between 68% and 82%) of the respondents could not demonstrate the correct calculations of the indicators. This is worrisome as it could pose a threat to quality of malaria indicators generated for decision-making in the MHD. There should be conscious efforts to build capacity of health staff on routine malaria indicators to improve their knowledge and competencies, as suggested by Ledikwe et al., [34]. Priority should be given to NM and CHN when it comes to training, since they constitute staff with the worst understanding and proficiency in the subject matter.

## Timeliness of routine malaria data

As indicated, the timeliness of submission of monthly report (including that of malaria) from the facility to the MHD is the 5th day of the ensuing month. Timeliness of reporting was below the target ($\geq$ 80%), which is not different from similar studies [34–36]. For instance, 6 out of 15 facilities were able to submit timely reports, which resulted a bit in delaying data entry and analysis at the MHD. This undoubtedly affects prompt decision-making regarding the malaria

elimination programme. The use of SMS for reporting malaria data has been identified as a promising practice for accurately tracking malaria trends. According to Yukich and colleagues, the rapid spread of this technology across Africa offers promising opportunities to collect and disseminate surveillance data in a timely way [36].

## Availability of routine malaria data

Availability and completeness of data, measure reporting performance to determine the extent to which data reports are appropriately available and complete. Each facility is supposed to submit a monthly malaria programme summary report to the MHD. As already established by Githinji et al., [37], our study affirms that overwhelming majority of the facilities had all their reports not only available but completed.

## Accuracy of malaria data

A person presenting with a history of fever within the preceding 2–3 days, or found to have fever on examination (axillary temperature 37.5˚C or rectal temperature 38.5˚C), in the absence of any other cause, will be considered a suspected case with malaria [37]. Almost half of the 15 facilities reported the exact number of suspected malaria cases that were clinically diagnosed and recorded in the Laboratory and Consulting Registers during the period of investigation. This makes their data appear to be of high quality and suitable for decision-making. Nevertheless, a sizeable number of the facilities did not over report or underreport their data, which is an indication of a good data for quality decision-making. Despite the splendid performance presented by 12 facilities, the remaining three over reported the real suspected cases, leading to poor data quality. That notwithstanding, the overall verification factor (VF) indicated the MHD had good data quality, which is a good indication that any collective decision-making based on this indicator is a reflection of reality.

Suspected malaria cases tested is defined as the occurrence of malaria illness/disease in a person in whom the presence of malaria parasites in the blood has been confirmed by parasitological testing. Again, our study shows that three out of every five facilities reported the exact number of suspected malaria cases recorded over the period, making it authentic for decision-making. However, one out of every five facilities did not over report or underreport their data, with moderate level of data quality which might affect the quality of its use. There were still some few facilities underreporting malaria cases tested. In spite of some discrepancies identified, the collaborative data looked good and suitable for decision-making. Number of malaria cases tested positive is defined as total number of suspected malaria cases that tested positive for malaria using microscopy or RDTs. Eight out of fifteen facilities reported the exact number of suspected malaria cases that tested positive in during lab investigations. This is a positive outcome as this might have led to sound decision on specific treatment regimen to reduce the burden of malaria episode. Another two out of every five facilities did not over report or underreported their data, which makes it good for decision-making. Even though one facility underreported, thus rendering their dataset as poor, that could not affect the collaborative MHD data which looked good and ingredient for effective decision-making.

Since 2004, it has been a national policy to use Artemisinin-based Combination Therapy (ACTs) for the treatment of uncomplicated malaria in Ghana [38]. This change was necessary because the malaria parasite became resistant to Chloroquine and other monotherapies. Artemisinin and its derivatives are the most rapidly acting and effective anti-malarias available. They are administered in combination with a second, long-acting anti-malaria drug in order to enhance treatment and protect against the development of drug resistance. Despite the importance of this indicator, none of the facilities reported the exact number of reported OPD

malaria cases treated with ACTs as noted by other researchers [36, 39]. Ironically, four out of every five facilities in the MHD over reported, rendering their data unusable, while the remaining had a moderate quality data. The collaborative effect of the poor level of quality exhibited by the facilities rendered the MHD data poor and useless for any meaningful decision-making. Considering the MHD verification factor, performances among the four individual malaria indicators: number of suspected malaria cases, number of suspected malaria cases tested, and number of malaria cases tested positive fell within the accepted margin plus or minus 10% of 100% of the global standard. However, the "Number of OPD malaria cases treated with ACTs" is outside the acceptable margins, indicating poor data (Fig 4). That notwithstanding, the MHD VF of malaria indicators (summative) is not over reported or underreported and therefore is good for making decisions related to malaria implementation programme in the MHD. Thus, the quality of malaria indicator in the MHD appears to be better contrary to what other studies found [36, 40, 41].

## Conclusions

The majority of staff had not received any training on malaria related health information management in the past six months prior to the study. This to a large extent has quality implications of malaria data produced at all levels of service delivery in the MHD. It was therefore not coincidence that some staff members did not know the correct definitions of some key indicators that were used in malaria programme, while the majority of them could not demonstrate the correct calculations of these indicators. This could pose a threat to quality of malaria indicators generated for evidence-based decision-making. There should be conscious efforts to build capacity of health staff on routine malaria indicators to improve their knowledge and competencies through training, and refresher training coupled with supportive supervisory visits to the facilities. Timeliness of reporting was below the target, as the majority of the facilities were unable to submit timely reports, which resulted in delayed in data entry and analysis at the MHD. This undoubtedly affected prompt decision-making regarding the malaria elimination programme. We propose the use of SMS for reporting malaria data to improve timeliness.

The MHD had their reports available, completed and accurate which was a good recipe for evidence based decision-making, pointing to improving malaria elimination programme. The four individual malaria indicators namely: number of suspected malaria cases, number of suspected malaria cases tested, and number of malaria cases tested positive performed within the accepted margin plus or minus 10% of 100% of the global standard. However, the "Number of OPD malaria cases treated with ACTs" were outside the acceptable margins, indicating poor data. We conclude that the VF of Malaria Indicators (summative) were not over reported or underreported and therefore were of quality (timeliness, availability, data accuracy) for making decisions related to malaria implementation programme in the MHD. Nonetheless, there should be continuous quality improvement focused training to enhance staff's proficiency and improved quality of key malaria indicators.

## Recommendation

We recommend interventions such as training, refresher training of frontline providers and strengthening of regular supportive supervision and monitoring by the MHD with technical and logistics support from NMCP. Routine malaria data generation at the health facilities may not be peculiar to malaria routine health information subsystems; but the reflection of a wider routine health information system (RHIS) weaknesses. Therefore, any interventions seeking to improve the system must look beyond just malaria health information subsystem initiatives

and include consideration of the broader contextual factors that improve RHIS. Notwithstanding, we again recommend for further research into why OPD malaria cases tested positive is not matching with cases treated with ACTs in health centres and clinics.

## Supporting information

**S1 File. Supporting information file containing the study data (sample size by each study arm, by study sites, total malaria cases tested, by time point etc.) and analysis.** (XLSX)

## Acknowledgments

Our special thanks go to the Sissala East Municipal Health Directorate especially the staff who supported in diverse ways towards the conduct of the study.

## Author Contributions

**Conceptualization:** Richard Okyere Boadu.

**Data curation:** Hor Karimeni Karimu.

**Formal analysis:** Richard Okyere Boadu, Hor Karimeni Karimu.

**Methodology:** Richard Okyere Boadu.

**Project administration:** Richard Okyere Boadu.

**Resources:** Richard Okyere Boadu.

**Supervision:** Richard Okyere Boadu.

**Writing – original draft:** Richard Okyere Boadu, Kwame Adu Okyere Boadu.

**Writing – review & editing:** Richard Okyere Boadu, Hor Karimeni Karimu, Kwame Adu Okyere Boadu, Obed Uwumbornyi Lasim, Lady Agyei Boatemaa, Solomon Abotiba Atinbire, Nathan Kumasenu Mensah.

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
