## [Decision Letter · Decision Letter 0]

2 May 2022

PONE-D-22-00045Assessment of health staffs’ proficiency and quality of key malaria indicators in Rural District of GhanaPLOS ONE

Dear Dr. Okyere Boadu,

Thank you for submitting your manuscript to PLOS ONE. After careful consideration, we feel that it has merit but does not fully meet PLOS ONE’s publication criteria as it currently stands. Therefore, we invite you to submit a revised version of the manuscript that addresses the points raised during the review process.

We look forward to receiving your revised manuscript.

Kind regards,

Pyae Linn Aung, Ph.D.

Academic Editor

PLOS ONE

Journal Requirements:

4. We note that you have referenced (Ghana Health Service, Sissala East District Municipal Annual Report (2020), Ghana Health Service, Sissala East District Municipal Annual Report (2019)) which has currently not yet been accepted for publication. Please remove this from your References and amend this to state in the body of your manuscript: (ie “Bewick et al. [Unpublished]”) as detailed online in our guide for authors http://journals.plos.org/plosone/s/submission-guidelines#loc-reference-style.

Additional Editor Comments:

The paper addressed an important issue regarding the health staff’s proficiency and quality of key malaria indicators in one of the malaria-endemic countries. I have some more concerns as below.

- For the selection of sites and participants, the authors selected 15 health facilities randomly, did you consider some other factors like workloads of each facility, staff distribution, the experience of staff, malaria caseloads, etc. between each facility?

- Include background of the health facility staff, e.g., what are the criteria to be a staff at different level, structure of each facility, organogram and detailed procedures and steps for the data entry, validating process, software if any, quality control and quality assurance of data, and so on.

- Interview with the staff was conducted. However, I do not see details of the questionnaire, or interviewing procedures (e.g., Who else perform the interview?).

- Results from the questionnaire were pooled (e.g., Chart 1). However, participants possessed different categories and positions together with separate educational attainment (Table 2). How did the authors control its impact? Could it be biased?

- Results section can be more concise by eliminating the repetitions between the words and contents in the tables.

- Overall, the analysis is a kind of thin and describes mainly descriptive. The sample size was rather small for the interview to calculate any inferential statistics.

- Are there any linkages between staff proficiency and the quality of key malaria indicators in each facility? Should there be a statistical test for this?

- A sentence is misleading under the recommendation section. The authors discussed NHIS. If it was not directly related to the objectives and outcomes of the study, for me I think the sentence should better be excluded.

- Ensure the references are followed Plos’s in-house styles.

- Recheck the long-form of LLIN (Page 2), spell out acronyms (e.g., ACTs, DQA on page 3) in the first appearance.

Reviewers' comments:

Reviewer's Responses to Questions

**Comments to the Author**

1. Is the manuscript technically sound, and do the data support the conclusions?

Reviewer #1: Yes

Reviewer #2: Yes

Reviewer #3: Yes

2. Has the statistical analysis been performed appropriately and rigorously? 

Reviewer #1: Yes

Reviewer #2: Yes

Reviewer #3: N/A

3. Have the authors made all data underlying the findings in their manuscript fully available?

Reviewer #1: No

Reviewer #2: No

Reviewer #3: No

4. Is the manuscript presented in an intelligible fashion and written in standard English?

Reviewer #1: Yes

Reviewer #2: Yes

Reviewer #3: No

5. Review Comments to the Author

Reviewer #1: The paper presents an assessment of the quality of some key malaria indicator reports from healthcare facilities in rural districts of Ghana. Results show that reports of major indicators are of good quality in terms of accuracy, completeness, and timeliness. However, there is one indicator whose reports have poor quality. The author concludes that the quality of the reports are good enough to support decision making.

The paper also accessed the proficiency of healthcare staff in using the malaria indicators. However, there seems not a direct link between the proficiency and the quality of the corresponding reports.

Reviewer #2: This study aims to investigate the data quality with respect to health staffs’ proficiency, timeliness, availability and data accuracy in the Sissala East MHD, based on a randomly collected dataset from 15 facilities and 50 health staffs in the area.

This paper is generally well written with reasonable experimental results and analysis, from which I believe the conclusion is solid. However, there are some minor issues:

1. In the abstract, the authors claim the data is collected from 24 August, 2020 – 17 September, 2020. However, in the main body of the paper, the data sources are tagger 2019, eg. Chart 2, Chart 3 and Chart 4.

2. The assessment is based on a dataset of 50 health staffs, which seems a bit small. It would be more convincing if more data is collected, or more details are provided, e.g., how many health staffs there are in each of the facilities.

3. Table 6 shows particularlly poor numbers. Is it because data incompleteness, i.e., with VF=0? I think corresponding explanation and discussion would help people better understand the table here.

Reviewer #3: Summary of the research and overall impression

This manuscript presented to identify the health staff’s proficiency and quality of data on key malaria indicators in selected rural district of Ghana. Overall, the study will benefit to the national malaria control program to understand the requirement and extent/ capacity of local health staff at the health facilities level. The findings will provide input for the national program to enhance and strengthen the data quality assurance which is one of the most important factors for malaria elimination.

Technically, data presentation and data analysis, methodology is good, however the major improvement is needed for presenting in text in the results session, discussion, conclusion and recommendation.

English language is strongly recommended for academic publication, in terms of using the correct tense (in verb), academic writing and language of data presentation throughout the manuscript.

Please refer the detail of the comments in the document uploaded.

6. PLOS authors have the option to publish the peer review history of their article (what does this mean?). If published, this will include your full peer review and any attached files.

Reviewer #1: No

Reviewer #2: No

Reviewer #3: **Yes: **Poe Poe Aung

---

## [Author Response · Author response to Decision Letter 0]

30 May 2022

On behalf of my colleagues, I am submitting responses to Reviewers and Editor’s comments raised in our article “Assessment of health staffs’ proficiency and quality of key malaria indicators in Rural District of Ghana”. 

The following areas of concern raised by the reviewers and Editor with responses (highlighted) as detailed below: 

Abstract

1. The information presented in the abstract is concise. However, the last two sentences of the result session are not clear. Please edit for clear statement. 

Thank you for your feedback. The sentences have been revised to read, “However, the verification factor (VF) of the overall malaria indicator shows…” [see lines 36-37]

2. In conclusion, it is needed to revise mentioning – what we conclude from the key findings and what is the major implication of the findings for the action plan or for the future. 

Thank you for your feedback. The conclusions have been revised to read, “Majority of staff had not received any training on malaria-related RHIS. Some staff do not know the correct definitions of some of MI used in malaria programmes while the majority of them could not demonstrate the correct calculations of MI. Timeliness of reporting was below the target, nevertheless, copies of data that were submitted were available and completed. There should be training, supervision and monitoring to enhance staffs’ proficiency and improved quality of MI.” [see lines 38-42]

3. Language editing is required for abstract. Academic writing should be edited and the past-tense (verb) should be used. 

Thank you for your feedback. Language editing has been done. 

Introduction 

4. Suggest to add “Global Technical Strategy for Malaria 2016–2030” in the list of reference. 

Thank you for your feedback. We have added to the reference list [see line 66]

1. Pg-2, paragraph-2, line-6: Please clarify “These facilities are mandated to use these standard tools to collect and present data”. 

Thank you for your feedback. The sentence has been rephrased to read, “GHS has therefore mandated the public and mission health facilities to use these registers and forms as standard tools to collect and present data” [see lines 71-72]

2. Pg-2, paragraph-3, line-6: Please expel “MICS”.

Thank you for your feedback. “MICS” is expelled from the text [see line 91]

3. Pg-2, paragraph-3, line-8: “28414 cases were confirmed” please mention confirmed by what method of diagnosis. (e.g. confirmed positive by RDT?) 

Thank you for your feedback. The sentence has been reconstructed to read, “For instance, about 28414 cases were confirmed positive by laboratory/RDT, …” [see line 94]

4. Pg-2, paragraph-3, line-12: Please correct the spelling of LLIN = “Long-lasting insecticidal nets” 

Thank you for your feedback. The error has been corrected [see line 97]

5. Pg-2, paragraph-4, line-10: “…..this is not different from other studies in Ghana and ……” please add the reference for this statement. The current reference [24] is from Uganda representing for “other sub-Saharan countries”. 

Thank you for your feedback. The original statement has been modified, and additional references added. The statement has been rephrased to read: “This has limited the use of malaria data to meet the reporting needs of the municipal. managers. Previous research in Ghana and other sub-Saharan African countries has highlighted issues of inaccurate reporting of malaria cases as well as treatment plans, wrongly categorizing malaria cases in registers used to collate data and misplacing data and or registers for reporting [25-28].” [see lines 107-110]

6. Lastly, please add more information for why this study is important to conduct (justification) and how this study can provide supplementary information or input to the health system, in addition to other studies about HMIS conducted in the Ghana. 

Thank you for your feedback. Additional information has been added while the sentence has been rephrased to read, “While previous attempts to address the issues of Malaria data quality has concentrated at the national level discrepancies with collated data against what pertains at district and regional levels. Little research has evaluated the contribution of facility-level staff proficiency in Malaria data quality to identify potential gaps and implement any data quality improvement intervention. This analysis is intending to contribute to filling this knowledge gap as Ghana moves towards the elimination of Malaria. This study presents findings of data quality concerning health staffs’ proficiency, timeliness, availability and data accuracy in the Sissala East MHD”. [see lines 111-116]

Methods

7. Study area: please include the total number of populations, male: female ratio, population density, total populations that one health facility covers, number of health staff (e.g. doctor: patient ratio, etc.)

Thank you for your feedback. Characteristics of the facilities included in the study are shown in Table 1. [see lines 137-142]

8. Selection of malaria indicator: please mention what type of malaria test was used for diagnosis and confirmed malaria positive (e.g. RDT, PCR, microscopy?)

Thank you for your feedback. Depending on the facility either microscopy or RDT or both tests were used for diagnosis and confirmed malaria positive. [see lines158-159]

Results 

9. Please add the data table and description of the background characteristics of study sites, study populations, etc. (e.g. sample size by each study arm, by study sites, totally tested, by time point etc.)

Thank you for your feedback. A data table and description of the background characteristics of study sites, and study populations are provided in Table 1[see lines 137-142]. An additional table showing sample size by each study arm, by study sites, total tested, time point etc. is provided as a supplementary material.

10. Expressing each indicator is good and comprehensive, but the description in the text for the table is too detail. Suggest only key findings should be presented in the text, not to duplicate information with the table and text. Please re-write the results text. 

Thank you for your feedback. Some sections of the results have been re-written [see lines 326-327, 335, 343, 349-351]

11. Table 2. The “total” row is not necessary. Please delete. Please re-order the category of staff according to the number from the largest to the smallest. 

Thank you for your feedback. Corrections have been made accordingly. [see lines 243-244]

12. Chart 4. Should explain and describe how the data in chart 4 was calculated, in terms of “summative” – which the term is only appeared and described in the discussion (one time) and conclusion (one time). 

Thank you for your feedback. Chart 4 has been explained and described as ‘The MHD VF of each indicator was summative of all “recounted” and “reported” data points from the fifteen selected health facilities. Also, the VF of the overall malaria indicator was summative of all “recounted” and “reported” data points for the key indicators (Number of suspected malaria cases, Number of suspected malaria cases tested, Number of malaria cases tested positive, Number of OPD malaria cases treated with ACTs from the fifteen selected health facilities’ [see lines 356-360]

Discussion and conclusions

13. Discussion: Please add more literature to compare and describe what are the difference in lessons learnt with other studies. The comparative of writing about other literature is weak in the discussion session. 

Thank you for your feedback. Literature regarding the selected key indicators for the study is scanty for comparative writing about other studies. However, we have managed to do a few more to enrich our discussions. [see lines 399-401]

14. Please discuss the point that VF values for the indicators (summative), except OPD cases treated with ACT, were within the acceptable range, but they all lied below “1” which is over-reported. Please explain and discuss for this point, which could tend to be more over-reported in the reality or other health facilities not included in the study. 

Thank you for your feedback. Our interpretation is consistent with the definitions provided in the methods section. Please refer to Table 2 [i.e. % of facilities that over-report by more than 10% (VF < 0.90, % of facilities that under-report by more than 10% (VF > 1.10)]

15. Conclusion: pg 14, line 6, it said malaria “eradication”. Is it truly that the program aimed for malaria eradication or elimination? Please recheck. 

Thank you for your feedback. The error has been corrected to “elimination”. [line 466] 

16. The conclusion should be more precise and concise, touching all key findings and recommendation for program implementation. There is a duplication of mentioning about the “OPD cases treated with ACT” and suggest to combine the duplicate description. Please revise the conclusion session. 

Thank you for your feedback. The conclusion has been revised to incorporate your feedback. [lines ….]

17. Recommendation must be training, refresher training and strengthen regular supervision & monitoring plan by the national program. Further research is required, but only focus on the OPD cases and ACT treatment indicator is not the key recommendation from this paper. Yes, it is the only one indicator which needs improvement and pay attention, but the recommendation should be more in general and benefit for the malaria program as an overview. Please revise the recommendation. 

Thank you for your feedback. The conclusion has been revised to incorporate your feedback. [lines 453-473]

Thank you for your reconsideration and re-evaluation prior to consideration for publication. My colleagues and I appreciate your time and effort and look forward to hearing from you.

---

## [Decision Letter · Decision Letter 1]

15 Jul 2022

PONE-D-22-00045R1Assessment of health staffs’ proficiency and quality of key malaria indicators in Rural District of GhanaPLOS ONE

Dear Dr. Okyere Boadu,

Thank you for submitting your manuscript to PLOS ONE. After careful consideration, we feel that it has merit but does not fully meet PLOS ONE’s publication criteria as it currently stands. Therefore, we invite you to submit a revised version of the manuscript that addresses the points raised during the review process.

We look forward to receiving your revised manuscript.

Kind regards,

Pyae Linn Aung, Ph.D.

Academic Editor

PLOS ONE

Additional Editor Comments:

Please revisit comments from the last round of review and carefully response each and every comment of editor and reviewers even if you want to push back the suggestions if any.

Reviewers' comments:

Reviewer's Responses to Questions

**Comments to the Author**

1. If the authors have adequately addressed your comments raised in a previous round of review and you feel that this manuscript is now acceptable for publication, you may indicate that here to bypass the “Comments to the Author” section, enter your conflict of interest statement in the “Confidential to Editor” section, and submit your "Accept" recommendation.

Reviewer #1: All comments have been addressed

Reviewer #2: All comments have been addressed

Reviewer #3: (No Response)

2. Is the manuscript technically sound, and do the data support the conclusions?

Reviewer #1: (No Response)

Reviewer #2: Yes

Reviewer #3: Yes

3. Has the statistical analysis been performed appropriately and rigorously? 

Reviewer #1: (No Response)

Reviewer #2: Yes

Reviewer #3: Yes

4. Have the authors made all data underlying the findings in their manuscript fully available?

Reviewer #1: (No Response)

Reviewer #2: No

Reviewer #3: Yes

5. Is the manuscript presented in an intelligible fashion and written in standard English?

Reviewer #1: (No Response)

Reviewer #2: Yes

Reviewer #3: No

6. Review Comments to the Author

Reviewer #1: (No Response)

Reviewer #2: It seems to me the authors have addressed the comments, and I therefore recommend acceptance of this paper.

Reviewer #3: Summary of the research and overall impression

The comments were addressed and revised accordingly. However, there are still required to improve in results and discussion sessions, in terms of technical data presentation and English language editing. I would strongly recommend to consult/ take the proper English language editing service for academic publication writing and language of data presentation throughout the manuscript.

Pls find the detail comments in the attached file.

7. PLOS authors have the option to publish the peer review history of their article (what does this mean?). If published, this will include your full peer review and any attached files.

Reviewer #1: No

Reviewer #2: No

Reviewer #3: **Yes: **Poe Poe Aung

---

## [Author Response · Author response to Decision Letter 1]

28 Jul 2022

The following areas of concern raised by the reviewers and Editor with responses (highlighted) as detailed below: 

Discussion on specific area of improvement 

1. Ethical consideration: pls mention the name of institution that the approval was issued, together with approval number. For example: Ethical approval was obtained from xxxx institution (approval number xxxx). 

Thank you for your feedback. The issues raised have been resolved. That is “Ethical approval was obtained from the Cape Coast Teaching Hospital Ethical Review Committee (approval number: CCTHERC/EC/2020/045)” [see lines 222-223]

Results 

2. Majority of the description in the text for the table and figures are still not revised yet. Pls describe and interpret the key findings ONLY in the text, not to duplicate information with the table and text. Pls revise all text in the result session and check the use of past-tense (verb) consistently in the result session in terms of English language editing. 

For example: “The specialization of respondents included: Doctor (2.0%), Pharmacist (2.0%),

252 Physician Assistant (2.0%), Nurses/Midwives (42.0%), Technical Officers (14.0%), Dispensing Technician (2.0%),253 Health Information Officers/ Biostatisticians (8.0%), and Lab scientist/Technician (6.0%) as shown in Table 3.” [need to revise, this is just description of the result and duplicated with the information from the table. This is not the interpretation of the key findings.]

The above sentence should be described as followed as the example. “The majority of the respondents were nurses/ midwives (42.0%), followed by the technical officers (14.0%) and health information officers/ biostatisticians (8.0%). Doctors and physician assistants were contributed only 2% of the respondents respectively.”

Thank you for your feedback. We have revised the results section and corrected all typo and grammatical errors that we could find. Furthermore, we had to rephrase certain constructions and dictions that seemed out of place [see lines 237-347]. The remaining tables 4-7 and their corresponding texts were not considered because we had already revised same, and for fear of run counter to whatever we wanted to communicate.

Discussion

3. Thank you for revising, yet, discussion is still needed to improve more concise in description in writing. I would recommend to consult/ take the proper English language editing service for academic writing. 

Thank you for your feedback. We have corrected all typo and grammatical errors that we could find. Furthermore, we had to rephrase certain constructions and dictions that seemed out of place [see lines 410-513].

---

## [Decision Letter · Decision Letter 2]

2 Sep 2022

Assessment of health staffs’ proficiency and quality of key malaria indicators in Rural District of Ghana

PONE-D-22-00045R2

Dear Dr. Okyere Boadu,

We’re pleased to inform you that your manuscript has been judged scientifically suitable for publication and will be formally accepted for publication once it meets all outstanding technical requirements.

Kind regards,

Pyae Linn Aung, Ph.D.

Academic Editor

PLOS ONE

Additional Editor Comments (optional):

Reviewers' comments:

Reviewer's Responses to Questions

**Comments to the Author**

1. If the authors have adequately addressed your comments raised in a previous round of review and you feel that this manuscript is now acceptable for publication, you may indicate that here to bypass the “Comments to the Author” section, enter your conflict of interest statement in the “Confidential to Editor” section, and submit your "Accept" recommendation.

Reviewer #3: All comments have been addressed

2. Is the manuscript technically sound, and do the data support the conclusions?

Reviewer #3: Yes

3. Has the statistical analysis been performed appropriately and rigorously? 

Reviewer #3: Yes

4. Have the authors made all data underlying the findings in their manuscript fully available?

Reviewer #3: Yes

5. Is the manuscript presented in an intelligible fashion and written in standard English?

Reviewer #3: Yes

6. Review Comments to the Author

Reviewer #3: The authors addressed all the comments and suggestion. No further comment and accept the revised version.

7. PLOS authors have the option to publish the peer review history of their article (what does this mean?). If published, this will include your full peer review and any attached files.

Reviewer #3: **Yes: **Poe Poe Aung

---

## [Editor Report · Acceptance letter]

7 Oct 2022

PONE-D-22-00045R2 

Assessment of health staff’s proficiency and quality of key malaria indicators in Rural District of Ghana 

Dear Dr. Okyere Boadu:

I'm pleased to inform you that your manuscript has been deemed suitable for publication in PLOS ONE. Congratulations! Your manuscript is now with our production department. 

Kind regards, 

on behalf of

Dr. Pyae Linn Aung 

Academic Editor

PLOS ONE